# Adaptive Concentration Inequalities for Sequential Decision Problems

**Shengjia Zhao**
Tsinghua University
zhaosj12@stanford.edu

**Enze Zhou**
Tsinghua University
zhouez_thu_12@126.com

**Ashish Sabharwal**
Allen Institute for AI
AshishS@allenai.org

**Stefano Ermon**
Stanford University
ermon@cs.stanford.edu

## Abstract

A key challenge in sequential decision problems is to determine how many samples are needed for an agent to make reliable decisions with good probabilistic guarantees. We introduce Hoeffding-like concentration inequalities that hold for a random, adaptively chosen number of samples. Our inequalities are tight under natural assumptions and can greatly simplify the analysis of common sequential decision problems. In particular, we apply them to sequential hypothesis testing, best arm identification, and sorting. The resulting algorithms rival or exceed the state of the art both theoretically and empirically.

## 1 Introduction

Many problems in artificial intelligence (AI) and machine learning (ML) involve designing agents that interact with stochastic environments. The environment is typically modeled with a collection of random variables. A common assumption is that the agent acquires information by observing samples from these random variables. A key problem is to determine the number of samples that are required for the agent to make sound inferences and decisions based on the data it has collected.

Many abstract problems fit into this general framework, including sequential hypothesis testing, e.g., testing for positiveness of the mean [18, 6], analysis of streaming data [19], best arm identification for multi-arm bandits (MAB) [1, 5, 13], etc. These problems involve the design of a sequential algorithm that needs to decide, at each step, either to acquire a new sample, or to terminate and output a conclusion, e.g., decide whether the mean of a random variable is positive or not. The challenge is that obtaining too many samples will result in inefficient algorithms, while taking too few might lead to the wrong decision.

Concentration inequalities such as Hoeffding's inequality [11], Chernoff bound, and Azuma's inequality [7, 5] are among the main analytic tools. These inequalities are used to bound the probability of a large discrepancy between sample and population means, for a fixed number of samples $n$. An agent can control its risk by making decisions based on conclusions that hold with high confidence, due to the unlikely occurrence of large deviations. However, these inequalities only hold for a fixed, *constant number of samples that is decided a-priori*. On the other hand, we often want to design agents that make decisions adaptively based on the data they collect. That is, we would like the *number of samples itself to be a random variable*. Traditional concentration inequalities, however, often do not hold when the number of samples is stochastic. Existing analysis requires ad-hoc strategies to bypass this issue, such as union bounding the risk over time [18, 17, 13]. These approaches can lead to suboptimal algorithms.

We introduce Hoeffding-like concentration inequalities that hold for a random, adaptively chosen number of samples. Interestingly, we can achieve our goal with a small double logarithmic overhead with respect to the number of samples required for standard Hoeffding inequalities. We also show that our bounds cannot be improved under some natural restrictions. Even though related inequalities have been proposed before [15, 2, 3], we show that ours are significantly tighter, and come with a complete analysis of the fundamental limits involved. Our inequalities are directly applicable to a number of sequential decision problems. In particular, we use them to design and analyze new algorithms for sequential hypothesis testing, best arm identification, and sorting. Our algorithms rival or outperform state-of-the-art techniques both theoretically and empirically.

## 2 Adaptive Inequalities and Their Properties

We begin with some definitions and notation:

**Definition 1.** [20] Let $X$ be a zero mean random variable. For any $d > 0$, we say $X$ is $d$-subgaussian if $\forall r \in \mathbb{R}$,

$$\mathbb{E}[e^{rX}] \leq e^{d^2 r^2 / 2}$$

Note that a random variable can be subgaussian only if it has zero mean [20]. However, with some abuse of notation, we say that any random variable $X$ is subgaussian if $X - \mathbb{E}[X]$ is subgaussian.

Many important types of distributions are subgaussian. For example, by Hoeffding's Lemma [11], a distribution bounded in an interval of width $2d$ is $d$-subgaussian and a Gaussian random variable $N(0, \sigma^2)$ is $\sigma$-subgaussian. Henceforth, we shall assume that the distributions are $1/2$-subgaussian. Any $d$-subgaussian random variable can be scaled by $1/(2d)$ to be $1/2$-subgaussian

**Definition 2** (Problem setup). Let $X$ be a zero mean $1/2$-subgaussian random variable. $\{X_1, X_2, \ldots\}$ are i.i.d. random samples of $X$. Let $S_n = \sum_{i=1}^{n} X_i$ be a random walk. $J$ is a stopping time with respect to $\{X_1, X_2, \ldots\}$. We let $J$ take a special value $\infty$ where $\Pr[J = \infty] = 1 - \lim_{n \to \infty} \Pr[J \leq n]$. We also let $f : \mathbb{N} \to \mathbb{R}^+$ be a function that will serve as a boundary for the random walk.

We note that because it is possible for $J$ to be infinity, to simplify notation, what we really mean by $\Pr[E_J]$, where $E_J$ is some event, is $\Pr[\{J < \infty\} \cap E_J]$. We can often simplify notation and use $\Pr[E_J]$ without confusion.

### 2.1 Standard vs. Adaptive Concentration Inequalities

There is a very large class of well known inequalities that bound the probability of large deviations by confidence that increases exponentially w.r.t. bound tightness. An example is the Hoeffding inequality [12] which states, using the definitions mentioned above,

$$\Pr[S_n \geq \sqrt{bn}] \leq e^{-2b} \tag{1}$$

Other examples include Azuma's inequality, Chernoff bound [7], and Bernstein inequalities [21]. However, these inequalities apply if $n$ is a constant chosen in advance, or independent of the underlying process, but are generally untrue when $n$ is a stopping time $J$ that, being a random variable, depends on the process. In fact we shall later show in Theorem 3 that we can construct a stopping time $J$ such that

$$\Pr[S_J \geq \sqrt{bJ}] = 1 \tag{2}$$

for any $b > 0$, even when we put strong restrictions on $J$.

Comparing Eqs. (1) and (2), one clearly sees how Chernoff and Hoeffding bounds are applicable only to algorithms whose decision to continue to sample or terminate is fixed a priori. This is a severe limitation for stochastic algorithms that have uncertain stopping conditions that may depend on the underlying process. We call a bound that holds for all possible stopping rules $J$ an *adaptive bound*.

### 2.2 Equivalence Principle

We start with the observation that finding a probabilistic bound on the position of the random walk $S_J$ that holds *for any stopping time $J$* is equivalent to finding a deterministic boundary $f(n)$ that the walk is unlikely to ever cross. Formally,

**Proposition 1.** *For any $\delta > 0$,*

$$\Pr[S_J \geq f(J)] \leq \delta \tag{3}$$

*for any stopping time $J$ if and only if*

$$\Pr[\{\exists n, S_n \geq f(n)\}] \leq \delta \tag{4}$$

Intuitively, for any $f(n)$ we can choose an adversarial stopping rule that terminates the process as soon as the random walk crosses the boundary $f(n)$. We can therefore achieve (3) for all stopping times $J$ only if we guarantee that the random walk is unlikely to ever cross $f(n)$, as in Eq. (4).

## 2.3  Related Inequalities

The problem of studying the supremum of a random walk has a long history. The seminal work of Kolmogorov and Khinchin [4] characterized the limiting behavior of a zero mean random walk with unit variance:

$$\limsup_{n \to \infty} \frac{S_n}{\sqrt{2n \log \log n}} = 1 \quad a.s.$$

This law is called the Law of Iterated Logarithms (LIL), and sheds light on the limiting behavior of a random walk. In our framework, this implies

$$\lim_{m \to \infty} \Pr\left[\exists n > m : S_n \geq \sqrt{2an \log \log n}\right] = \begin{cases} 1 & \text{if } a < 1 \\ 0 & \text{if } a > 1 \end{cases}$$

This theorem provides a very strong result on the asymptotic behavior of the walk. However, in most ML and statistical applications, we are also interested in the *finite-time behavior*, which we study.

The problem of analyzing the finite-time properties of a random walk has been considered before in the ML literature. It is well known, and can be easily proven using Hoeffding's inequality union bounded over all possible times, that a trivial bound

$$f(n) = \sqrt{n \log(2n^2/\delta)/2} \tag{5}$$

holds in the sense of $\Pr\left[\exists n, S_n \geq f(n)\right] \leq \delta$. This is true because by union bound and Hoeffding inequality [12]

$$Pr[\exists n, S_n \geq f(n)] \leq \sum_{n=1}^{\infty} Pr[S_n \geq f(n)] \leq \sum_{n=1}^{\infty} e^{-\log\left(2n^2/\delta\right)} \leq \delta \sum_{n=1}^{\infty} \frac{1}{2n^2} \leq \delta$$

Recently, inspired by the Law of Iterated Logarithms, Jamieson et al. [15], Jamieson and Nowak [13] and Balsubramani [2] proposed a boundary $f(n)$ that scales asymptotically as $\Theta(\sqrt{n \log \log n})$ such that the "crossing event" $\{\exists n, S_n \geq f(n)\}$ is guaranteed to occur with a low probability. They refer to this as finite time LIL inequality. These bounds, however, have significant room for improvement. Furthermore, [2] holds asymptotically, i.e., only w.r.t. the event $\{\exists n > N, S_n \geq f(n)\}$ for a sufficiently large (but finite) $N$, rather than across all time steps. In the following sections, we develop general bounds that improve upon these methods.

## 3  New Adaptive Hoeffding-like Bounds

Our first main result is an alternative to finite time LIL that is both tighter and simpler:

**Theorem 1** (Adaptive Hoeffding Inequality). *Let $X_i$ be zero mean $1/2$-subgaussian random variables. $\{S_n = \sum_{i=1}^{n} X_i, n \geq 1\}$ be a random walk. Let $f : \mathbb{N} \to \mathbb{R}^+$. Then,*

1. *If $\lim_{n \to \infty} \frac{f(n)}{\sqrt{(1/2)n \log \log n}} < 1$, there exists a distribution for $X$ such that*

$$\Pr[\{\exists n, S_n \geq f(n)\}] = 1$$

2. *If $f(n) = \sqrt{an \log(\log_c n + 1) + bn}$, $c > 1$, $a > c/2$, $b > 0$, and $\zeta$ is the Riemann-$\zeta$ function, then*

$$\Pr[\{\exists n, S_n \geq f(n)\}] \leq \zeta\left(2a/c\right) e^{-2b/c} \tag{6}$$

We also remark that in practice the values of $a$ and $c$ do not significantly affect the quality of the bound. We recommend fixing $a = 0.6$ and $c = 1.1$ and will use this configuration in all subsequent experiments. The parameter $b$ is the main factor controlling the confidence we have on the bound (6), i.e., the risk. The value of $b$ is chosen so that the bound holds with probability at least $1 - \delta$, where $\delta$ is a user specified parameter.

Based on Proposition 1, and fixing $a$ and $c$ as above, we get a readily applicable corollary:

**Corollary 1.** *Let $J$ be any random variable taking value in $\mathbb{N}$. If*

$$f(n) = \sqrt{0.6n \log(\log_{1.1} n + 1) + bn}$$

*then*

$$\Pr[S_J \geq f(J)] \leq 12e^{-1.8b}$$

The bound we achieve is very similar in form to Hoeffding inequality (1), with an extra $O(\log \log n)$ slack to achieve robustness to stochastic, adaptively chosen stopping times. We shall refer to this inequality as the Adaptive Hoeffding (AH) inequality.

Informally, part 1 of Theorem 1 implies that if we choose a boundary $f(n)$ that is convergent w.r.t. $\sqrt{n \log \log n}$ and would like to bound the probability of the threshold-crossing event, $\sqrt{(1/2)n \log \log n}$ is the asymptotically smallest $f(n)$ we can have; anything asymptotically smaller will be crossed with probability 1. Furthermore, part 2 implies that as long as $a > 1/2$, we can choose a sufficiently large $b$ so that threshold crossing has an arbitrarily small probability. Combined, we thus have that for any $\kappa > 0$, the minimum $f$ (call it $f^*$) needed to ensure an arbitrarily small threshold-crossing probability can be bounded asymptotically as follows:

$$\sqrt{1/2}\sqrt{n \log \log n} \leq f^*(n) \leq (\sqrt{1/2} + \kappa)\sqrt{n \log \log n} \qquad (7)$$

This fact is illustrated in Figure 1, where we plot the bound $f(n)$ from Corollary 1 with $12e^{-1.8b} = \delta = 0.05$ (AH, green). The corresponding Hoeffding bound (red) that would have held (with the same confidence, had $n$ been a constant) is plotted as well. We also show draws from an unbiased random walk (blue). Out of the 1000 draws we sampled, approximately 25% of them cross the Hoeffding bound (red) before time $10^5$, while none of them cross the adaptive bound (green), demonstrating the necessity of the extra $\sqrt{\log \log n}$ factor even in practice.

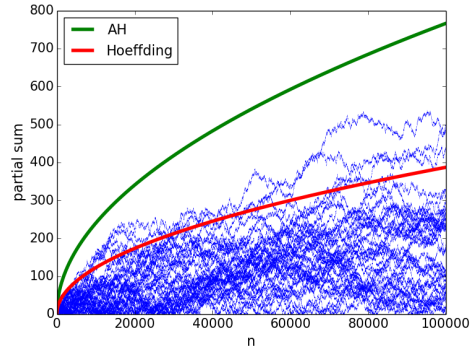

We also compare our bound with the trivial bound (5), LIL bound in Lemma 1 of [15] and Theorem 2 of [2]. The graph in Figure 2 shows the relative performance of the three bounds across different values of $n$ and risk $\delta$. The LIL bound of [15] is plotted with parameter $\epsilon = 0.01$ as recommended. We also experimented with other values of $\epsilon$, obtaining qualitatively similar results. It can be seen that our bound is significantly tighter (by roughly a factor of 1.5) across all values of $n$ and $\delta$ that we evaluated.

Figure 1: Illustration of Theorem 1 part 2. Each blue line represents a sampled walk. Although the probability of reaching higher than the Hoeffding bound (red) at a given time is small, the threshold is crossed almost surely. The new bound (green) remains unlikely to be crossed.

### 3.1 More General, Non-Smooth Boundaries

If we relax the requirement that $f(n)$ must be smooth, or, formally, remove the condition that

$$\lim_{n \to \infty} \frac{f(n)}{\sqrt{n \log \log n}}$$

must exist or go to $\infty$, then we might be able to obtain tighter bounds.

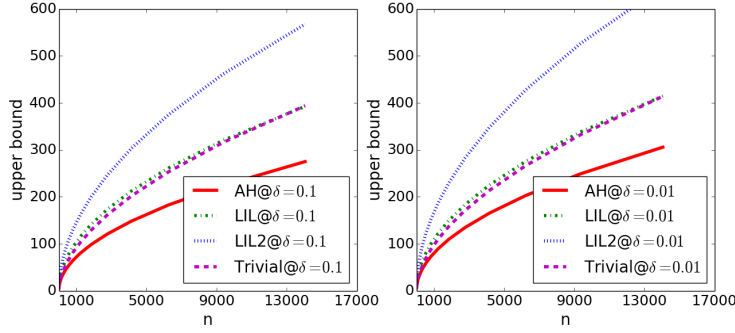

Figure 2: Comparison of Adaptive Hoeffding (AH) and LIL [15], LIL2 [2] and Trivial bound. A threshold function $f(n)$ is computed and plotted according to the four bounds, so that crossing occurs with bounded probability $\delta$ (risk). The two plots correspond to different risk levels (0.01 and 0.1).

For example many algorithms such as median elimination [9] or the exponential gap algorithm [17, 6] make (sampling) decisions "in batch", and therefore can only stop at certain pre-defined times. The intuition is that if more samples are collected between decisions, the failure probability can be easier to control. This is equivalent to restricting the stopping time $J$ to take values in a set $\mathfrak{N} \subset \mathbb{N}$. Equivalently we can also think of using a boundary function $f(n)$ defined as follows:

$$f_{\mathfrak{N}}(n) = \begin{cases} f(n) & n \in \mathfrak{N} \\ +\infty & otherwise \end{cases} \tag{8}$$

Very often the set $\mathfrak{N}$ is taken to be the following set:

**Definition 3** (Exponentially Sparse Stopping Time). We denote by $\mathfrak{N}_c$, $c > 1$, the set $\mathfrak{N}_c = \{\lceil c^n \rceil : n \in \mathbb{N}\}$.

Methods based on exponentially sparse stopping times often achieve asymptotically optimal performance on a range of sequential decision making problems [9, 18, 17]. Here we construct an alternative to Theorem 1 based on exponentially sparse stopping times. We obtain a bound that is asymptotically equivalent, but has better constants and is often more effective in practice.

**Theorem 2** (Exponentially Sparse Adaptive Hoeffding Inequality). *Let $\{S_n, n \geq 1\}$ be a random walk with $1/2$-subgaussian increments. If*

$$f(n) = \sqrt{an \log(\log_c n + 1) + bn}$$

*and $c > 1$, $a > 1/2$, $b > 0$, we have*

$$\Pr[\{\exists n \in \mathfrak{N}_c, S_n \geq f(n)\}] \leq \zeta(2a)\, e^{-2b}$$

We call this inequality the exponentially sparse adaptive Hoeffding (ESAH) inequality. Compared to (6), the main improvement is the lack of the constant $c$ in the RHS. In all subsequent experiments we fix $a = 0.55$ and $c = 1.05$.

Finally, we provide limits for any boundary, including those obtained by a batch-sampling strategy.

**Theorem 3.** *Let $\{S_n, n \geq 1\}$ be a zero mean random walk with $1/2$-subgaussian increments. Let $f : \mathbb{N} \to \mathbb{R}^+$. Then*

1. *If there exists a constant $C \geq 0$ such that $\liminf_{n \to \infty} \frac{f(n)}{\sqrt{n}} < C$, then*

$$\Pr[\{\exists n, S_n \geq f(n)\}] = 1$$

2. *If $\lim_{n \to \infty} \frac{f(n)}{\sqrt{n}} = +\infty$, then for any $\delta > 0$ there exists an infinite set $\mathfrak{N} \subset \mathbb{N}$ such that*

$$\Pr[\{\exists n \in \mathfrak{N}, S_n \geq f(n)\}] < \delta$$

Informally, part 1 states that if a threshold $f(n)$ drops an infinite number of times below an asymptotic bound of $\Theta(\sqrt{n})$, then the threshold will be crossed with probability 1. This rules out Hoeffding-like bounds. If $f(n)$ grows asymptotically faster than $\sqrt{n}$, then one can "sparsify" $f(n)$ so that it will be crossed with an arbitrarily small probability. In particular, a boundary with the form in Equation (8) can be constructed to bound the threshold-crossing probability below any $\delta$ (part 2 of the Theorem).

## 4 Applications to ML and Statistics

We now apply our adaptive bound results to design new algorithms for various classic problems in ML and statistics. Our bounds can be used to analyze algorithms for many natural sequential problems, leading to a unified framework for such analysis. The resulting algorithms are asymptotically optimal or near optimal, and outperform competing algorithms in practice. We provide two applications in the following subsections and leave another to the appendix.

### 4.1 Sequential Testing for Positiveness of Mean

Our first example is sequential testing for the positiveness of the mean of a bounded random variable. In this problem, there is a $1/2$-subgaussian random variable $X$ with (unknown) mean $\mu \neq 0$. At each step, an agent can either request a sample from $X$, or terminate and declare whether or not $\mathbb{E}[X] > 0$. The goal is to bound the agent's error probability by some user specified value $\delta$.

This problem is well studied [10, 18, 6]. In particular Karp and Kleinberg [18] show in Lemma 3.2 ("second simulation lemma") that this problem can be solved with an $O\left(\log(1/\delta)\log\log(1/\mu)/\mu^2\right)$ algorithm with confidence $1 - \delta$. They also prove a lower bound of $\Omega\left(\log\log(1/\mu)/\mu^2\right)$. Recently, Chen and Li [6] referred to this problem as the SIGN-$\xi$ problem and provided similar results.

We propose an algorithm that achieves the optimal asymptotic complexity and performs very well in practice, outperforming competing algorithms by a wide margin (because of better asymptotic constants). The algorithm is captured by the following definition.

**Definition 4** (Boundary Sequential Test). Let $f : \mathbb{N} \to \mathbb{R}^+$ be a function. We draw i.i.d. samples $X_1, X_2, \ldots$ from the target distribution $X$. Let $S_n = \sum_{i=1}^{n} X_i$ be the corresponding partial sum.

1. If $S_n \geq f(n)$, terminate and declare $\mathbb{E}[X] > 0$;

2. if $S_n \leq -f(n)$, terminate and declare $\mathbb{E}[X] < 0$;

3. otherwise increment $n$ and obtain a new sample.

We call such a test a *symmetric boundary test*. In the following theorem we analyze its performance.

**Theorem 4.** *Let $\delta > 0$ and $X$ be any $1/2$-subgaussian distribution with non-zero mean. Let*

$$f(n) = \sqrt{an\log(\log_c n + 1) + bn}$$

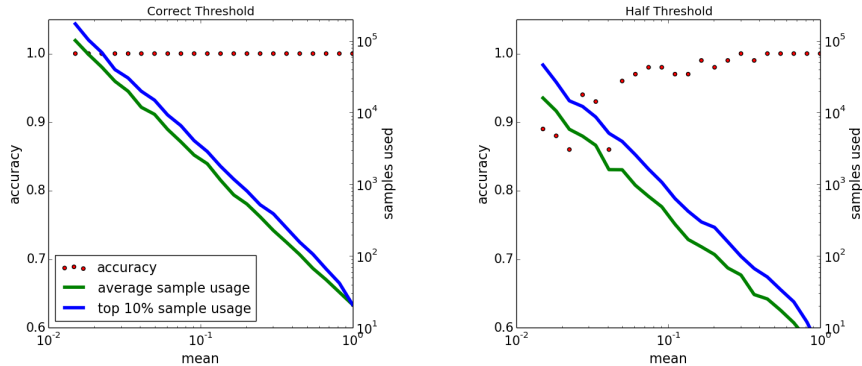

Figure 3: Empirical Performance of Boundary Tests. The plot on the left is the algorithm in Definition 4 and Theorem 4 with $\delta = 0.05$, the plot on the right uses half the correct threshold. Despite of a speed up of 4 times, the empirical accuracy drops below the requirement

*where $c > 1$, $a > c/2$, and $b = c/2 \log \zeta (2a/c) + c/2 \log 1/\delta$. Then, with probability at least $1 - \delta$, a symmetric boundary test terminates with the correct sign for $\mathbb{E}[X]$, and with probability $1 - \delta$, for any $\epsilon > 0$ it terminates in at most*

$$(2c + \epsilon) \left( \frac{\log(1/\delta) \log \log (1/\mu)}{\mu^2} \right)$$

*samples asymptotically w.r.t. $1/\mu$ and $1/\delta$.*

### 4.1.1 Experiments

To evaluate the empirical performance of our algorithm (AH-RW), we run an experiment where $X$ is a Bernoulli distribution over $\{-1/2, 1/2\}$, for various values of the mean parameter $\mu$. The confidence level $\delta$ is set to $0.05$, and the results are averaged across 100 independent runs. For this experiment and other experiments in this section, we set the parameters $a = 0.6$ and $c = 1.1$. We plot in Figure 3 the empirical accuracy, average number of samples used (runtime), and the number of samples after which 90% of the runs terminate.

The empirical accuracy of AH-RW is very high, as predicted by Theorem 4. Our bound is empirically very tight. If we decrease the bound by a factor of 2, that is we use $f(n)/2$ instead of $f(n)$, we get the curve in the right hand side plot of Figure 3. Despite a speed up of approximately 4 times, the empirical accuracy gets below the 0.95 requirement, especially when $\mu$ is small.

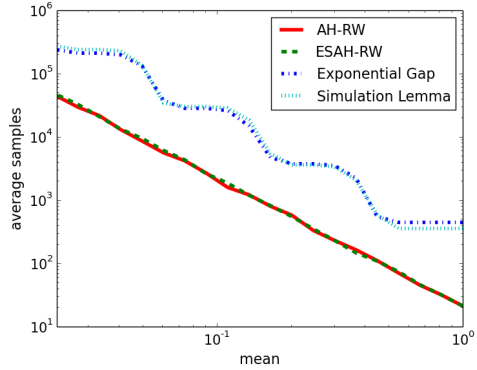

We also compare our method, AH-RW, to the Exponential Gap algorithm from [6] and the algorithm from the "second simulation lemma" of [18]. Both of these algorithms rely on a batch sampling idea and have very similar performance. The results show that our algorithm is at least an order of magnitude faster (note the log-scale). We also evaluate a variant of our algorithm (ESAH-RW) where the boundary function $f(n)$ is taken to be $f_{\mathfrak{N}_c}$ as in Theorem 2 and Equation (8). This algorithm achieves very similar performance as Theorem 4, justifying the practical applicability of batch sampling.

Figure 4: Comparison of various algorithms for deciding the positiveness of the mean of a Bernoulli random variable. AH-RW and ESAH-RW use orders of magnitude fewer samples than alternatives.

## 4.2 Best Arm Identification

The MAB (Multi-Arm Bandit) problem [1, 5] studies the optimal behavior of an agent when faced with a set of choices with unknown rewards. There are several flavors of the problem. In this paper, we focus on the fixed confidence best arm identification problem [13]. In this setting, the agent is presented with a set of arms $\mathbb{A}$, where the arms are indistinguishable except for their expected reward. The agent is to make sequential decisions at each time step to either pull an arm $\alpha \in \mathbb{A}$, or to terminate and declare one arm to have the largest expected reward. The goal is to identify the best arm with a probability of error smaller than some pre-specified $\delta > 0$.

To facilitate the discussion, we first define the notation we will use. We denote by $K = |\mathbb{A}|$ as the total number of arms. We denote by $\mu_\alpha$ the true mean of an arm, $\alpha^* = \arg \max \mu_\alpha$, We also define $\hat{\mu}_\alpha(n_\alpha)$ as the empirical mean after $n_\alpha$ pulls of an arm.

This problem has been extensively studied, including recently [8, 14, 17, 15, 6]. A survey is presented by Jamieson and Nowak [13], who classify existing algorithms into three classes: action elimination based [8, 14, 17, 6], which achieve good asymptotics but often perform unsatisfactorily in practice; UCB based, such as lil'UCB by [15]; and LUCB based approaches, such as [16, 13], which achieve sub-optimal asymptotics of $O(K \log K)$ but perform very well in practice. We provide a new algorithm that out-performs all previous algorithm, including LUCB, in Algorithm 1.

**Theorem 5.** *For any $\delta > 0$, with probability $1 - \delta$, Algorithm 1 outputs the optimal arm.*

**Algorithm 1** Adaptive Hoeffding Race (set of arms $\mathbb{A}$, $K = |\mathbb{A}|$, parameter $\delta$)

---

fix parameters $a = 0.6, c = 1.1, b = c/2 \left(\log \zeta \left(2a/c\right) + \log(2/\delta)\right)$

initialize for all arms $\alpha \in \mathbb{A}$, $n_\alpha = 0$, initialize $\hat{\mathbb{A}} = \mathbb{A}$ be the set of remaining arms

**while** $\hat{\mathbb{A}}$ has more than one arm **do**

    Let $\hat{\alpha}^*$ be the arm with highest empirical mean, and compute for all $\alpha \in \hat{\mathbb{A}}$

$$f_\alpha(n_\alpha) = \begin{cases} \sqrt{\left(a \log(\log_c n_\alpha + 1) + b + c \log |\hat{\mathbb{A}}|/2\right)/n_\alpha} & \text{if } \alpha = \hat{\alpha}^* \\ \sqrt{\left(a \log(\log_c n_\alpha + 1) + b\right)/n_\alpha} & \text{otherwise} \end{cases}$$

    draw a sample from the arm with largest value of $f_\alpha(n_\alpha)$ from $\hat{\mathbb{A}}$, $n_\alpha = n_\alpha + 1$

    remove from $\hat{\mathbb{A}}$ arm $\alpha$ if $\hat{\mu}_a + f_\alpha(n_\alpha) < \hat{\mu}_{\hat{\alpha}^*} - f_{\hat{\alpha}^*}(n_{\hat{\alpha}^*})$

**end while**

return the only element in $\hat{\mathbb{A}}$

---

### 4.2.1 Experiments

We implemented Algorithm 1 and a variant where the boundary $f$ is set to $f_{\mathfrak{N}_c}$ as in Theorem 2. We call this alternative version ES-AHR, standing for exponentially sparse adaptive Hoeffding race. For comparison we implemented the lil'UCB and lil'UCB+LS described in [14], and lil'LUCB described in [13]. Based on the results of [13], these algorithms are the fastest known to date.

We also implemented the DISTRIBUTION-BASED-ELIMINATION from [6], which theoretically is the state-of-the-art in terms of asymptotic complexity. Despite this fact, the empirical performance is orders of magnitude worse compared to other algorithms for the instance sizes we experimented with.

We experimented with most of the distribution families considered in [13] and found qualitatively similar results. We only report results using the most challenging distribution we found

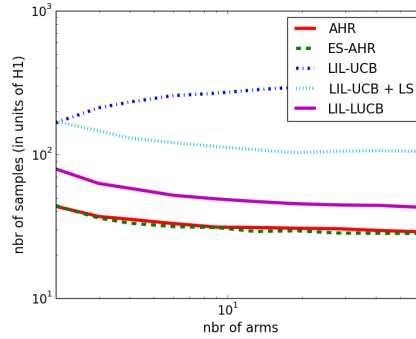

Figure 5: Comparison of various methods for best arm identification. Our methods AHR and ES-AHR are significantly faster than state-of-the-art. Batch sampling ES-AHR is the most effective one.

that was presented in that survey, where $\mu_i = 1 - (i/K)^{0.6}$. The distributions are Gaussian with 1/4 variance, and $\delta = 0.05$. The sample count is measured in units of $H_1 = \sum_{\alpha \neq \alpha^*} \Delta_\alpha^{-2}$ hardness [13].

## 5 Conclusions

We studied the threshold crossing behavior of random walks, and provided new concentration inequalities that, unlike classic Hoeffding-style bounds, hold for any stopping rule. We showed that these inequalities can be applied to various problems, such as testing for positiveness of mean, best arm identification, obtaining algorithms that perform well both in theory and in practice.

### Acknowledgments

This research was supported by NSF (#1649208) and Future of Life Institute (#2016-158687).

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
