[Supplementary Material]

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

# A Appendix

## A.1 Equivalence Principle

To show the correctness of Proposition 1, we first show and prove a more general lemma

**Lemma 1.** *Let $\mathfrak{F} = \{\mathcal{F}_n\}$ be a filtration for some discrete stochastic process. Let $\mathbb{I}_n$ be the indicator function for some event sequence measurable w.r.t. $\mathfrak{F}$. For any $\delta > 0$,*

$$\Pr[\mathbb{I}_J = 1] \leq \delta$$

*for any random variable $J$ taking values in $\mathbb{N}$ if and only if*

$$\Pr[\{\exists n, \mathbb{I}_n = 1\}] \leq \delta$$

*Proof of Lemma 1.* To prove sufficiency we show that for any sample path

$$\mathbb{I}_J \leq \sup_{n \in \mathbb{N}} \mathbb{I}_n = \mathbb{I}\{\exists n, \mathbb{I}_n = 1\}$$

Therefore

$$\Pr[\mathbb{I}_J = 1] \leq \Pr[\mathbb{I}\{\exists n, \mathbb{I}_n = 1\} = 1]$$
$$= \Pr[\{\exists n, \mathbb{I}_n = 1\}]$$

which implies $\Pr[\mathbb{I}_J = 1] \leq \delta$ if $\Pr[\{\exists n, \mathbb{I}_n = 1\}] \leq \delta$.

To show the necessity, we construct the following stopping time random variable

$$J = \inf\{n \in \mathbb{N} : \mathbb{I}_n = 1\}$$

Then for any sample path

$$\mathbb{I}\{\exists n, \mathbb{I}_n = 1\} \leq \mathbb{I}_J$$

which as before imply

$$\Pr[\{\exists n, \mathbb{I}_n = 1\}] \leq \Pr[\mathbb{I}_J = 1]$$

which implies that if $\Pr[\{\exists n, \mathbb{I}_n = 1\}] \leq \delta$ does not hold, there is a stopping time $J$ for which $\Pr[\mathbb{I}_J = 1] \leq \delta$ does not hold. □

*Proof of Proposition 1.* Follows from Lemma 1, taking the indicator function to be

$$\mathbb{I}_n = \mathbb{I}\{S_n \geq f(n)\}$$

□

## A.2 Proof of adaptive concentration inequalities

To prove the adaptive concentration inequalities we establish a lemma that will be used in the proof.

**Lemma 2.** *Let $X$ and $\{X_n, n \geq 1\}$ be a sequence of i.i.d $1/2$-subgaussian zero mean random variables. Let $S_n = \sum_{i=1}^{n} X_i$ be a zero mean random walk. Then*

$$\Pr\left[\max_{1 \leq i \leq n} S_i \geq \sqrt{an}\right] \leq e^{-2a}$$

*holds for all $a > 0$, $n \geq 1$*

*Proof of Lemma 2.* First we remark that because the distribution is subgaussian, the moment generating function $M_X(r)$ for $X$ exists for all values of $r$. Therefore $M_{S_n}(r)$ for $S_n$ also exists for all values of $n$ and $r$, because

$$M_{S_n}(r) = (M_X(r))^n$$

The sequence $\{S_n, n \geq 1\}$ is a martingale. We can apply a variant of Bernstein inequality for submartingales [21] (which can be proved from Kolmogorov's submartingale inequality) to have, for any $\alpha > 0$ and $r > 0$, and wherever $M_{S_n}(r)$ exists,

$$\Pr\left[\max_{1 \leq i \leq n} S_i \geq n\alpha\right] \leq M_{S_n}(r)e^{-rn\alpha}$$

Because the increments are $1/2$-subgaussian

$$M_X(r) \leq e^{r^2/8}$$

We must have

$$M_{S_n}(r) \leq e^{nr^2/8}$$

Combined we have, for all $r > 0$

$$\Pr\left[\max_{1 \leq i \leq n} S_i \geq n\alpha\right] \leq e^{nr^2/8}e^{-rn\alpha}$$

$$= e^{-n(r\alpha - r^2/8)}$$

The RHS is minimized when $\phi(r) = r\alpha - r^2/8$ is maximized. We can take the derivative of this term and setting to zero we have

$$\phi'(r) = \alpha - 2r/8 = 0$$

which leads to

$$r = 4\alpha$$

Therefore the tightest version of the original inequality is

$$\Pr\left[\max_{1 \leq i \leq n} S_i \geq n\alpha\right] \leq e^{-2n\alpha^2}$$

This can be viewed as a strengthened version of the original Hoeffding inequality [11] by upper bounding $\max_{1 \leq i \leq n} S_i$ rather than $S_n$ only. We can proceed to make $\alpha = \sqrt{a/n}$ and have

$$\Pr\left[\max_{1 \leq i \leq n} S_i \geq \sqrt{an}\right] \leq e^{-2a}$$

$\square$

*Proof of Theorem 1.* We first prove part 1 of the theorem. We can construct the distribution $X = 1/2$ with probability $1/2$ and $X = -1/2$ with probability $1/2$. This distribution is $1/2$-subgaussian, and the standard deviation $\sigma(X) = 1/2$. Recall that $X_i$ are i.i.d. samples from $X$. By the law of iterated logarithms [4], if $X_n$ has non-zero variance $\sigma^2$, $X_n/\sigma$ has unit variance and

$$\limsup_{n \to \infty} \frac{S_n}{\sigma\sqrt{2n \log \log n}} = 1 \qquad a.s$$

where $a.s$ indicates that the event has probability measure 1, or happens "almost surely". This means that for the distribution we constructed

$$\limsup_{n \to \infty} \frac{S_n}{\sqrt{1/2n \log \log n}} = 1 \qquad a.s$$

If $\lim_{n \to \infty} \frac{f(n)}{\sqrt{1/2n \log \log n}} < 1$, for any $\epsilon > 0$ and $N > 0$, so that $\forall n > N$

$$f(n) < (1 - \epsilon)\sqrt{1/2n \log \log n}$$

and regardless of our choice of $N$, almost surely there exists $n \geq N$ so that

$$\frac{S_n}{\sqrt{1/2n \log \log n}} > 1 - \epsilon > \frac{f(n)}{\sqrt{1/2n \log \log n}}$$

which implies $S_n \geq f(n)$. We next prove the second part of the theorem. Suppose we choose a monotonic non-decreasing $f(n)$, and $c > 1$.

$$\Pr[\{\exists n, S_n \geq f(n)\}] = \Pr\left[\cup_{n=1}^{\infty}\{S_n \geq f(n)\}\right]$$

$$= \Pr\left[\cup_{l=0}^{\infty} \cup_{c^l \leq n \leq c^{l+1}} \{S_n \geq f(n)\}\right]$$

$$\leq \Pr\left[\cup_{l=0}^{\infty}\left\{\max_{c^l \leq n \leq c^{l+1}} S_n \geq f(c^l)\right\}\right] \qquad (9)$$

$$\leq \sum_{l=0}^{\infty} \Pr\left[\max_{1 \leq n \leq c^{l+1}} S_n \geq f(c^l)\right]$$

where 9 is derived from the monotonicity of $f(n)$ and the last step is by union bound. We take $f(n) = \sqrt{an \log(\log_c n + 1) + bn}$, which is indeed monotonic non-decreasing and apply Lemma 2 to obtain

$$\Pr[\{\exists n, S_n \geq f(n)\}] \leq \sum_{l=0}^{\infty} \Pr\left[\max_{1 \leq n \leq c^{l+1}} S_n \geq \sqrt{\left(\frac{a \log(l+1) + b}{c}\right)c^{l+1}}\right]$$

$$\leq \sum_{l=0}^{\infty} e^{-\frac{2a}{c}\log(l+1)}e^{-2b/c} = \sum_{l=0}^{\infty}(l+1)^{-2a/c}e^{-2b/c}$$

$$= \sum_{l=1}^{\infty} l^{-2a/c}e^{-2b/c}$$

$$= \zeta\left(\frac{2a}{c}\right)e^{-2b/c}$$

$\square$

*Proof of Theorem 2.* Proof of Theorem 2 is essentially the same as that of Theorem 1:

$$\Pr[\{\exists n \in \mathfrak{N}_c, S_n \geq f(n)\}] = \Pr\left[\bigcup_{n \in \mathfrak{N}_c}\{S_n \geq f(n)\}\right]$$

$$\leq \sum_{n \in \mathfrak{N}_c} \Pr[S_n \geq f(n)]$$

Again taking $f(n) = \sqrt{an \log(\log_c n + 1) + bn}$ and apply Hoeffding Inequality [12] we have

$$\Pr[\{\exists n \in \mathfrak{N}_c, S_n \geq f(n)\}] \leq \sum_{n \in \mathfrak{N}_c} e^{-2a \log(\log_c n+1)-2b}$$

$$\leq \sum_{l=0}^{\infty}(l+1)^{-2a}e^{-2b}$$

$$= \zeta(2a)e^{-2b}$$

$\square$

*Proof of Theorem 3.* We denote by $E_n$ the event $\{S_n \geq C\sqrt{n}\}$ and $\bar{E}_n$ as its complement. Consider the probability of $E_n|S_m = s_m$ for some $n > m$. Because of the memoryless property of the random walk

$$Pr\{S_n \geq C\sqrt{n}|S_m = s_m\} = Pr\{S_{n-m} \geq C\sqrt{n} - s_m\}$$

For any constant $D > C$, there exists $M(D) \geq 0$, so that for all $n \geq M(D)$, we have

$$C\sqrt{n} - s_m \leq D\sqrt{n}$$

Because $S_{n-m} \geq D\sqrt{n}$ would imply $S_{n-m} \geq C\sqrt{n} - s_m$, which means for such $n \geq M(D)$,

$$Pr\{S_n \geq C\sqrt{n}|S_m = s_m\} = Pr\{S_{n-m} \geq C\sqrt{n} - s_m\} \geq Pr\{S_{n-m} \geq D\sqrt{n}\} \quad (10)$$

By the central limit theorem (CLT), the distribution function $F_n(x)$ for $\frac{S_n}{\sigma\sqrt{n}}$, where $\sigma$ is the standard deviation of $X_i$, converges to the standard normal $\mathcal{N}(0,1)$ with distribution function $\Phi(x)$, or alternatively,

$$\lim_{n \to \infty} S_n/\sqrt{n} \to_D \mathcal{N}(0, \sigma^2)$$

Where $\to_D$ indicates convergence in distribution. Because for all sample path $S_m/\sqrt{n} \to 0$, by Fatou's lemma,

$$\lim_{n \to \infty} S_m/\sqrt{n} \to_p 0$$

where $\to_p$ denote convergence in probability. By Theorem2.7 in [22] we have

$$\lim_{n \to \infty} (S_n - S_m)/\sqrt{n} \to_D \mathcal{N}(0, \sigma^2)$$

therefore
$$\lim_{n \to \infty} \Pr[S_{n-m}/\sqrt{n} \geq D] = \lim_{n \to \infty} \Pr[(S_n - S_m)/\sqrt{n} \geq D] = 1 - \Phi(D/\sigma) \qquad (11)$$
Combining 10 and 11 we have for $n \geq M(D)$,
$$\Pr[E_n \mid S_m = s_m] \geq 1 - \Phi(D/\sigma) > 0$$
Therefore for a sequence of integer time steps $n > m_1 > \cdots > m_k$,
$$\lim_{n \to \infty} \Pr[E_n \mid \bar{E}_{m_1}, ..., \bar{E}_{m_k}]$$
$$= \lim_{n \to \infty} \sum_{s_{m_1}} \Pr[E_n \mid S_{m_1} = s_{m_1}, \bar{E}_{m_1}, ..., \bar{E}_{m_k}] \Pr[S_{m_1} = s_{m_1} \mid \bar{E}_{m_1}, ..., \bar{E}_{m_k}]$$
$$\geq \sum_{s_{m_1}} (1 - \Phi(D/\sigma)) \Pr[S_{m_1} = s_{m_1} \mid \bar{E}_{m_1}, ..., \bar{E}_{m_k}]$$
$$= 1 - \Phi(D/\sigma) > 0$$
So there exists some $\epsilon > 0$ and a function $\mathcal{N} : \mathbb{N} \to \mathbb{N}$, so that for any $m_1 \geq 0$, and any $\forall n > \mathcal{N}(m_1)$,
$$\Pr[E_n \mid \bar{E}_{m_1}, ..., \bar{E}_{m_k}] \geq \epsilon$$
regardless of the choice for $m_2, \cdots m_k$.

If $\liminf_{n \to \infty} \frac{f(n)}{\sqrt{n}} < C$, then there is any infinite ordered sequence $Q = \{Q_0, Q_1, ...\} \subset \mathbb{N}^+$ so that $Q_0 > 0$, and $\forall Q_i \in Q, Q_{i+1} > \mathcal{N}(Q_i)$ and
$$f(Q_i) \leq C\sqrt{Q_i}$$
Then
$$\Pr[\{\nexists n, S_n \geq f(n)\}] \leq \Pr\left[\bigcap_{n \in Q} \bar{E}_n\right]$$
$$= \prod_{i \in \mathbb{N}} \Pr\left[\bar{E}_{Q_i} \mid \bar{E}_{Q_{i-1}}, ..., \bar{E}_{Q_0}\right]$$
$$\leq \prod_{i \in \mathbb{N}} (1 - \epsilon) = 0$$

Now we prove the second part. By Hoeffding Inequality [12],
$$Pr[S_n \geq f(n)] \leq e^{-2f^2(n)/n}$$
which means
$$\lim_{n \to \infty} \Pr[S_n \geq f(n)] = 0$$
under our assumption $\lim_{n \to \infty} \frac{f(n)}{\sqrt{n}} = \infty$. We call this probability $q(n) = \Pr[S_n \geq f(n)]$. We construct a sequence $Q \subset \mathbb{N}^+$ indexed by $i \in \mathbb{N}^+$ recursively as follows:
$$Q_1 = \inf_{n \in \mathbb{N}} \left\{n : q(n) < \frac{\delta}{2}\right\}$$
$$Q_i = \inf_{n \in \mathbb{N}} \left\{n > Q_{i-1} \mid q(n) < \frac{\delta}{2^i}\right\}$$
If is easy to see that Q can be constructed because $\lim_{n \to \infty} q(n) = 0$, and we can always find sufficiently large $n$ so that $q(n) < \delta/2^i, \forall i \in \mathbb{N}^+$. Furthermore $Q$ is an infinite monotonic increasing sequence by definition. Therefore
$$\Pr[\{\exists n \in Q, S_n \geq f(n)\}] = \Pr\left[\bigcup_{n \in Q} \{S_n \geq f(n)\}\right]$$
$$\leq \sum_{n \in Q} q(n) \leq \delta \sum_{i=1}^{\infty} \frac{1}{2^i} = \delta$$

$\square$

### A.3 Proof for sequential testing for positiveness of mean

To prove Theorem 4 we first prove a lemma

**Lemma 3.** *If we let $\mu, \delta, a, b > 0$, $c > 1$, and*

$$f(n) = \sqrt{an \log(\log_c n + 1) + bn}$$

*Define*

$$J = \inf\{n : f(n) \leq \mu n\}$$

*Then asymptotically as $\mu \to 0$,*

$$J \leq (a+b)\frac{\log \log (1/\mu)}{\mu^2}$$

*Proof of Lemma 3.* In the following we often neglect lower order terms. When we do so we use $\sim$ rather than $=$, and $\lesssim$ rather than $\leq$. The alternatives carry the same meaning, only that they hold as $\mu \to 0$

We first define $I = \gamma \frac{\log \log (1/\mu)}{\mu^2}$, where $\gamma$ is some constant we will later fix. Our proof strategy is to show that if $\gamma$ satisfies $\gamma > a + b$, asymptotically $f(I) \leq \mu I$, which makes $I$ an upper bound for $J$.

$$\log_c(I) = \log_c \gamma + \log_c \log \log \frac{1}{\mu} + 2\log_c \frac{1}{\mu} \sim \frac{2}{\log c} \log \frac{1}{\mu}$$

and

$$\log(\log_c(I) + 1) \sim \log \log \frac{1}{\mu}$$

Therefore, neglecting low order terms we have,

$$f(I) \sim \sqrt{\gamma \frac{\log \log(1/\mu)}{\mu^2}\left(a \log \log \frac{1}{\mu} + b\right)}$$

$$\lesssim \sqrt{\gamma(a+b)}\frac{\log \log (1/\mu)}{\mu} \sim \sqrt{\frac{a+b}{\gamma}}\mu I$$

Because we only neglected lower order terms

$$\lim_{\mu \to 0} \frac{f(I)}{\mu I} \leq \sqrt{\frac{a+b}{\gamma}}$$

For $f(I) \leq \mu I$ asymptotically, it suffices to have

$$\sqrt{\frac{a+b}{\gamma}} < 1$$

or

$$\gamma > a + b$$

Therefore $I$ constitutes an upper bound for $J$ whenever $\gamma > a + b$ $\qquad \square$

*Proof of Theorem 4.* Without loss of generality, we assume $E[X] = \mu > 0$, and let $S'_n = S_n - \mu n$ be a new zero mean random walk on the same sample space.

To prove the correctness we note that the algorithms terminates incorrectly only if $\exists n, S_n \leq -f(n)$, or equivalently $\exists n, S'_n \leq -f(n) - \mu n$. By Theorem 1

$$Pr[\exists n \in \mathbb{N}, S'_n \leq -f(n) - \mu n] \leq Pr[\exists n \in \mathbb{N}, S'_n \leq -f(n)] = Pr[\exists n \in \mathbb{N}, -S'_n \geq f(n)] \leq \delta$$

To bound the runtime we show by Theorem 1, for any stopping time $J$ taking values in $\mathbb{N}$, with probability at least $1 - \delta$, we have

$$S_J \geq S'_J \geq -f(J)$$

For any value for $J$ that satisfies
$$\mu J \geq 2f(J)$$
with probability at least $1 - \delta$
$$f(J) \leq -f(J) + \mu J \leq S'_J + \mu J = S_J$$
and the termination criteria $S_J \geq f(J)$ is met. So with probability at least $1 - \delta$, the algorithm terminates within
$$\inf\{J : \mu J \geq 2f(J)\}$$
steps. By Lemma 3, this is satisfied asymptotically as $\mu \to 0$ and $\delta \to 0$ when
$$J \lesssim 4\left(a + \frac{c}{2}\log\zeta\left(\frac{2a}{c}\right) + \frac{c}{2}\log\frac{1}{\delta}\right)\frac{\log\log(1/\mu)}{\mu^2} \lesssim (2c + \epsilon)\frac{\log(1/\delta)\log\log(1/\mu)}{\mu^2}$$
where $\epsilon > 0$ is any constant. $\qquad\qquad\square$

## A.4  Proof for best arm identification

*Proof of Theorem 5.* To avoid notational confusion, we first remark that the set $\mathbb{A}$ changes as arms are eliminated, and we use $\mathbb{A}_0$ to denote the original arm set.

To prove the correctness of the algorithm we define $E_n$ as the event that the optimal arm is eliminated at time step $n$. We define a random variable $J$ to be the time step either the algorithm terminates, or eliminates the optimal arm, whichever happens first. $J$ is a well defined random variable because for any sample path $J$ takes a deterministic value in $\mathbb{N}$. We denote $J_\alpha$ as the number of pulls made to each arm at time step $J$, and $\hat{\mathbb{A}}_J$ as the set of available (not eliminated) arms. For $E_J$ to happen, there must be $\alpha \neq \alpha^* \in \hat{\mathbb{A}}_J$, so that $\hat{\mu}_\alpha$ is the maximum out of all empirical means, and
$$\hat{\mu}_{\alpha^*} + f_{\alpha^*}(J_{\alpha^*}) < \hat{\mu}_\alpha - f_\alpha(J_\alpha)$$
We let the event of an underestimation upon termination $U_J(\alpha)$ be the event
$$U_J(\alpha) = \{\hat{\mu}_\alpha(J_\alpha) + f_\alpha(J_\alpha) < \mu_\alpha\}$$
and the event of an overestimation $O_J(\alpha)$ as the event
$$O_J(\alpha) = \{\alpha = \arg\max_{\alpha\in\hat{\mathbb{A}}_J}\hat{\mu}_\alpha(J_\alpha)\} \cap \{\hat{\mu}_\alpha - f_\alpha(J_\alpha) > \mu_\alpha\}$$
Noting that
$$f_\alpha(n_\alpha) \geq \sqrt{(a\log(\log_c n_\alpha + 1) + b)/n_\alpha}$$
and because $n_\alpha(\hat{\mu}_\alpha(n_\alpha) - \mu_\alpha)$ is a zero mean random walk, by Theorem 1 and Lemma 1 we can have
$$Pr[U_J(\alpha)] \leq Pr\left[\hat{\mu}_\alpha(J_\alpha) - \mu_\alpha \leq -\sqrt{(a\log(\log_c J_\alpha + 1) + b)/J_\alpha}\right] \leq \frac{\delta}{2}$$
and similarly
$$\Pr[O_J(\alpha)] \leq \zeta\left(\frac{2a}{c}\right)e^{-2b/c-2c\log|\hat{\mathbb{A}}_J|/(2c)} \leq \frac{\delta}{2|\hat{\mathbb{A}}_J|}$$
If neither $U_J(\alpha^*)$ nor $O_J(\alpha), \forall\alpha \neq \alpha^*$ happen, then $\bar{U}_J(\alpha^*) \cap \bigcap_{\alpha\neq\alpha^*}\bar{O}_J(\alpha)$ implies for any $\alpha \neq \alpha^*$, either
$$\alpha \neq \arg\max_{\alpha\in\hat{\mathbb{A}}_J}\hat{\mu}_\alpha(J_\alpha)$$
or
$$\hat{\mu}_{\alpha^*} + f_{\alpha^*}(J_{\alpha^*}) \geq \mu_{\alpha^*} > \mu_\alpha \geq \hat{\mu}_\alpha - f_\alpha(J_\alpha)$$
and $E_J$ (best arm is eliminated) cannot happen. Therefore
$$E_J \subset U_J(\alpha^*) \cup \left(\bigcup_{\alpha\neq\alpha^*\in\mathbb{A}_J} O_J(\alpha)\right)$$
Which means that
$$\Pr[E_J] \leq \delta/2 + (|\hat{\mathbb{A}}_J| - 1)\frac{\delta}{2|\hat{\mathbb{A}}_J|} \leq \delta$$

Because $J$ is the time step the algorithm terminates, or eliminates the optimal arm, which ever happens first. Therefore for any sample path where $E_J$ does not happen, the algorithm never eliminated the optimal arm, including during the final iteration. Therefore the algorithm must terminate correctly. The set of such sample paths has probability measure at least $1 - \delta$.

$\square$