[Reviews · NeurIPS 2016]

Reviewer 1

Summary

This paper transforms the Law of the Iterated Logarithm into a Hoeffding-style bound for situations with N not fixed in advance but determined by a stopping rule. Compared to earlier attempts, the bound is noticeably tighter, and the authors illustrate its usefulness on sequential testing and on a Hoeffding Race algorithm for best arm identification.

Qualitative Assessment

Paper contains important results. I did not find any flaws (but I only checked the proof of main result, Theorem 1, in detail). The paper is also very well written: rather than just providing an abstract theorem, the authors illustrate the strength of the bound (and the fact that Hoeffding bound doesn't hold with non-fixed N, and the difference to earlier bounds) with very clear simulations. They also show how the bound really matters in practically relevant problems. A clear accept for me. Minor issues: - I would like to see some explanation why your bound is better than the previous ones by Jamieson and collaborators (for the bound by Balsubramani this is clear). - Also would be nice to have some more explanation of why 'a and c do not significantly affect the quality of the bound', as you state. What exactly do you mean by this? What if c is, say 20? - Definition 4: Bounary -> Boundary.

Confidence in this Review

2-Confident (read it all; understood it all reasonably well)


Reviewer 2

Summary

This paper introduces Hoeffding-like concentration inequalities that hold a random, adaptively chosen number of samples. The new inequality can be used to design and analyze new algorithms for sequential hypothesis testing etc. The basic idea is to consider a stopped process, S_J, instead of the regular process S_n with fixed index. There are existing theory and methods in probability to bound the stopped process. But I think the aspect of using this for the studying learning problems, such as sequential testing and multi-arm bandit problem, is a new and interesting perspective. The derivations are fairly standard. But I think it introduces (or reminds) useful tools to this community.

Qualitative Assessment

The experiments are preliminary. It would be nice to have more demonstration for wider range of applications of this new tool as mentioned in the paper.

Confidence in this Review

2-Confident (read it all; understood it all reasonably well)


Reviewer 3

Summary

For stopping time based random sums of centred subgaussian random variables (enough thro' 1/2-subgaussian rvs) Hoeffding type high probability bounds are presented (AHI, Th 1). ESAH for a given low probability, is via a suitably sparsely sampling when the threshold f(\cdot) asymptotically grows faster than \sqrt{n}. As applications of their theory, a sequential procedure to test the sign of mean a random variable and bandit-MDP are presented (but, sorting is also mentioned towards the end of Sec 1 as well as in Conclusions, but, presented?). These results improve the existing results.

Qualitative Assessment

I believe the results are important and will have impact. Perhaps a typo in the last expression (estimate for # of samples required) of Th 4: \mu is the unknown mean of the X and can be -ve; the numerator has log(\frac{1}{\mu}); is it \mu^2 (as in denominator)? Minor comment: A spelling typo in Def 4.

Confidence in this Review

2-Confident (read it all; understood it all reasonably well)


Reviewer 4

Summary

The paper derives adaptive Hoeffding-like concentration inequality for any stopping time. While the conventional Hoeffding-like concentration inequality holds only for the case in which the sample size n is a fixed variable, the paper gives a much stronger result on the case in which the sample size can be also random. It gives a theoretical results on the new concentration inequality and gives applications of the bound to two problems in machine learning and statistics: sequential testing and multi-armed bandit problem.

Qualitative Assessment

The motivation of this paper is clear and the results are strong in my opinion. That is, the paper gives the theoretical results on the new inequality and clearly shows the benefit of it to the application of statistics and machine learning. It is shown that algorithms derived from the new bound can outperform previous state-of-the-art algorithms for the considered problems. I believe that the paper is strong enough for the publication in NIPS.

Confidence in this Review

2-Confident (read it all; understood it all reasonably well)


Reviewer 5

Summary

The authors obtain Hoeffding type inequalities for stopped sequences. The classical Hoeffding inequality is a decaying exponential bound on the probability of a rare events for a sum of iid random variables. But this inequality is valid for a fixed number of random variables. The authors claim that it is of interest to obtain similar inequalities for random number of random variables. The authors appear to succeed in obtaining such an inequality (Thm 1 and Cor 1). The bound is valid for subgaussian random variables. The authors then claim that they can use this inequality and the idea behind it to obtain competing algorithms for some popular problems in statistics and machine learning (MAB etc, Thm 5).

Qualitative Assessment

The paper is very well written, is easy to follow, and the contributions are stated clearly and the problem is well motivated. I think the paper is a good contribution to the literature. I do have few comments that the authors may use to improve the paper. 1. Algorithm 1: The way it is written makes the concepts confusing. f_\alpha is defined such that its value depends on the knowledge of alpha^*. The latter is not known. So, how do you "draw a sample from the arm with largest value of f_\alpha", and also how do you implement the next line: remove from A ...? You cannot compare the inequality because alpha^* is not known. Perhaps the authors have made a mistake here in terms of notations? 2. Defintion 4 should be stated as an Algorithm? 3. The authors may wish to add few lines on how their inequalities obtained are not just a rewording of the law of iterated logarithm (LIL), or is not just a simple implication. I mean if you provide some intuition on the nontrivially of these results given LIL, it would make the paper complete.

Confidence in this Review

1-Less confident (might not have understood significant parts)